# Applications of Smart Water Management Systems: A Literature Review

**Érico Soares Ascenção ***, **Fernando Melo Marinangelo** , **Carlos Frederico Meschini Almeida** , **Nelson Kagan** and **Eduardo Mário Dias**

Department of Electrical Energy and Automation Engineering, Polytechnic School of the University of São Paulo, São Paulo 05508-010, SP, Brazil; femarinangelo@hotmail.com (F.M.M.); cfmalmeida@usp.br (C.F.M.A.); nelson.kagan@usp.br (N.K.); emdias@pea.usp.br (E.M.D.)

**\*** Correspondence: erico.ascencao@usp.br

**Abstract:** Issues such as climate change, water scarcity, population growth, and distribution losses have stimulated the use of new technologies to manage water resources. This is how the concept of smart water management emerged as a subcategory of the concept of smart cities. This article aimed first to identify the applications of smart water-management systems described in academic articles either as applications in development or as applications already implemented or as future trends; and, second, to classify them according to the processes in the value chain of public water supply services. To this end, a systematic review of the literature was carried out, in which 100 mentions of applications were identified in 62 selected articles; then, the mentions were grouped into 10 categories. The most frequent application categories were smart meters, implementation models and architectures, and loss management. Among the processes of the value chain, applications in processes of distribution and water use were highly predominant. The lack of detail about the integration between the different applications for a smart water-management system was pointed out as a limitation and an opportunity for future research development, especially in terms of a technological roadmap study based on the relationship between smart meters and loss management.

**Keywords:** smart water network; smart meter; water-management systems architecture; loss management; water consumption monitoring for users; prediction of water demand



## 1. Introduction

Water plays a fundamental role in people's lives, from the simple fact that it makes up more than half of the human body to its application in various sectors such as healthcare, food production, agriculture, and industry [1]. The management of water resources, that is, "the activity of planning, developing, distributing, and managing the optimal use of water resources" [2] (p. 4), has been facing different challenges in the last several years: climate change, droughts, water scarcity, distribution losses, population growth, infrastructure problems, and lack of user awareness [3].

Given this scenario, the discussion about the sustainability of water supply systems gained strength [4]. The world is increasingly looking for solutions to face problems related to the management of water resources, such as monitoring and controlling losses in distribution [5], predicting users' demand [6], and water quality [7]. In this respect, the adoption of new technologies is increasing, which is denoted by the emergence of the concept of Water 4.0 coined by the German Water Partnership, meaning the implementation of Industry 4.0 technologies in the sanitation sector [8]. Furthermore, a survey conducted between 2013 and 2014 on applications of smart metering technologies in water supply systems in Australia and New Zealand [9] showed that 250,000 smart meters were already installed or in the planning stage of being installed and 66% of the water service providers interviewed considered this aspect in their plans for the following 12 months.

Smart water management is a subcategory of the smart city concept [3,10–16]. Neirotti et al. [17] defined smart water management as one of the application domains of smart cities that had lower coverage rates in the cities analyzed in the study in question; however, they did not detail the applications within the concept of smart water networks. Also, no other review article regarding smart water-management systems applications was found. This context led to the following research questions:

- Q1: What are the applications of smart water-management systems presented in the academic literature?
- Q2: How can those applications cover the value chain of public water supply services?

The objective of this paper was, first, to identify the applications of smart water-management systems described in academic articles either as applications in development or as applications already implemented or as future trends; and, second, to classify them according to the processes in the value chain of public water supply services.

## 2. Literature Review

### 2.1. Smart Water Networks

Smart water networks are water supply systems that mix physical and virtual elements in order to enable better system performance and resilience, as well as greater user engagement [18]. Benítez et al. [19] defined this concept in a more tangible way, by characterizing smart water networks as those that have a large number of devices for measuring variables such as pressure, flow, and totalized flow in an automatic and continuous way (physical elements); and, also, mentioning how the data from these meters (virtual elements) could be used, for example, for loss detection and prediction of demand.

Günther et al. [20] (p. 452) treated the concept of smart water networks more deeply, describing them as "the functional interaction of different components installed in a water distribution system" and unraveling the physical and virtual aspects in the following layers: physical, measurement and control, data collection and communication, data monitoring, and data analysis. The physical layer includes the physical infrastructure of the water network, i.e., pipes, reservoirs, valves, pumps, tanks, etc. The measurement and control layer is composed of field measurement instruments (flow, pressure, reservoir level, and water-quality parameters such as turbidity, conductivity, and free chlorine; for example) and control devices such as programmable logic controllers (PLCs). The data collection and communication layer includes SCADA (supervisory control and data acquisition) systems, the telemetry infrastructure, and the data itself transmitted over the communication network. The data management and monitoring layer is responsible for storing, processing, and making this data available to interested parties. Finally, the data-analysis layer consists of a set of tools that use data for different applications, which can be categorized into functionalities.

### 2.2. Smart Water Management

Smart water management has been treated in the literature as one of the application domains of smart cities, along with smart grids, urban mobility, garbage collection, citizen services, healthcare, and safety [10,16]. Tzagkarakis et al. [21] correlated the "intelligence" in question with the search for greater service efficiency in a smart city. Similarly, Simmhan et al. [16] and Slaný et al. [22] defined smart water management as the use of internet of things (IoT) technologies with the following objectives: guarantee the quality and availability of water, avoid losses, preventively maintain the infrastructure (networks, equipment, etc.), and engage users to save water. Therefore, it can be inferred that smart water management consists in how to use smart water networks to deliver a sustainable, efficient, and quality service to the end user.

### 2.3. Water Value Chain

In the same way that smart water management constitutes one of the application domains within the concept of smart cities, it can also be subdivided into application

domains. The value-chain concept, then, becomes an auxiliary tool. According to [23,24], a value chain is a representation of a company's processes in a detailed way, denoting how it delivers its final product to the customer. Therefore, from the analysis of the water value chain, it is possible to identify the application domains used as references to analyze the coverage of applications of smart water-management systems.

The Basic Sanitation Company of the State of São Paulo (Sabesp) defines in its Sustainability Report [25] the value chain illustrated in Figure 1. Sabesp is the largest sanitation company in the Americas and the fifth largest in the world, in terms of population served, and responsible for serving 28 million people in 375 municipalities in the São Paulo state (Brazil).

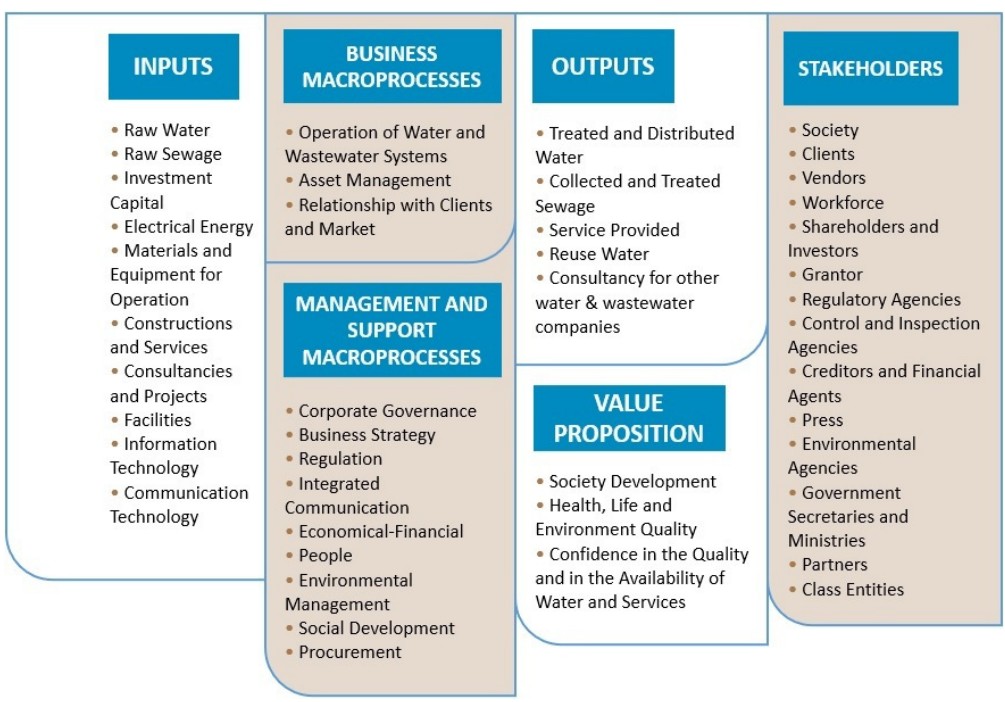

**Figure 1.** Value chain defined by Sabesp (adapted from [25]).

The water value chain has a strong relationship with the water cycle in nature. In its analysis of the processes applied for water reuse, Prisciandaro et al. [26] propose a closed water cycle, considering water capture, treatment, storage, and distribution processes in addition to sewage treatment and effluent reuse (see Figure 2). The water cycle proposed by these authors represented more thoroughly and comprehensively the main processes identified by [25] and, for this reason, it was adopted as a reference for the analysis of the coverage of smart water-management system applications in the water value chain.

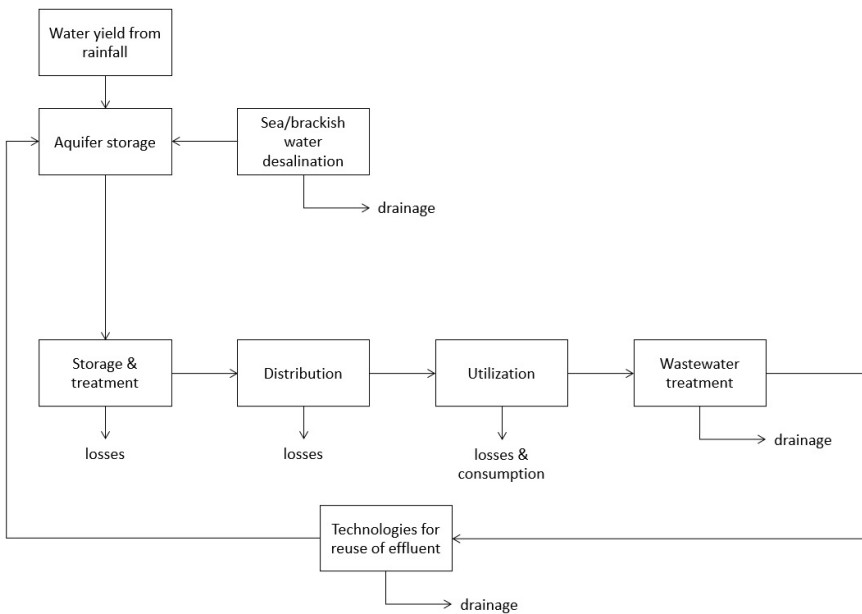

**Figure 2.** Closed water cycle (adapted from [26]).

## 3. Methods

The methodology adopted in this paper was that of a systematic literature review illustrated in Figure 3 using the PRISMA framework [27], which searched for articles dealing with smart water-management systems and their applications. The initial article search was carried out on the Web of Science database platform on March 22, 2022, using the following search string:

- *(TI = ("smart water management")) OR (TI = ("smart water network")) OR (TI = ("smart water grid")) OR (TI = ("smart water meter")) OR (TI = ("smart water sensor")) OR (AB = ("smart water management")) OR (AB = ("smart water network")) OR (AB = ("smart water grid")) OR (AB = ("smart water meter")) OR (AB = ("smart water sensor")) OR (AK = ("smart water management")) OR (AK = ("smart water network")) OR (AK = ("smart water grid")) OR (AK = ("smart water meter")) OR (AK = ("smart water sensor"))*

This search string resulted in a set of 212 articles. Next, review articles, abstracts, and editorial material were excluded from the set, so that only articles published in scientific journals, conferences, and congress annals remained, including the "early access" ones. In the end, the set comprised 194 articles.

Three criteria for the selection of articles to be reviewed were defined: (a) minimum of ten citations, (b) publication from 2019 with at least one citation, and (c) minimum average of one annual citation. The criteria were regressively restrictive (37, 58, and 73 selected articles, respectively). Finally, we decided to select the articles which met at least one of the three criteria, resulting in a total sample of 98 articles.

The abstracts of the selected articles were analyzed independently by the authors to identify applications of smart water-management systems. The analyzed articles were entered in the lines of a Microsoft Excel spreadsheet and the applications in the columns; then, the applications mentioned in each article were highlighted. During the analysis stage, 36 articles were excluded because they either focused on other research areas (agriculture, energy, rainwater drainage, and sectorization of supply networks) or did not address applications related to smart water-management systems; therefore, 62 articles remained. A total of 100 application mentions were identified and grouped into 10 categories by similarity.

Finally, the selected articles were classified according to the processes of the value chain to which their mentions of applications referred, allowing the crossing of application categories with the processes of the value chain.

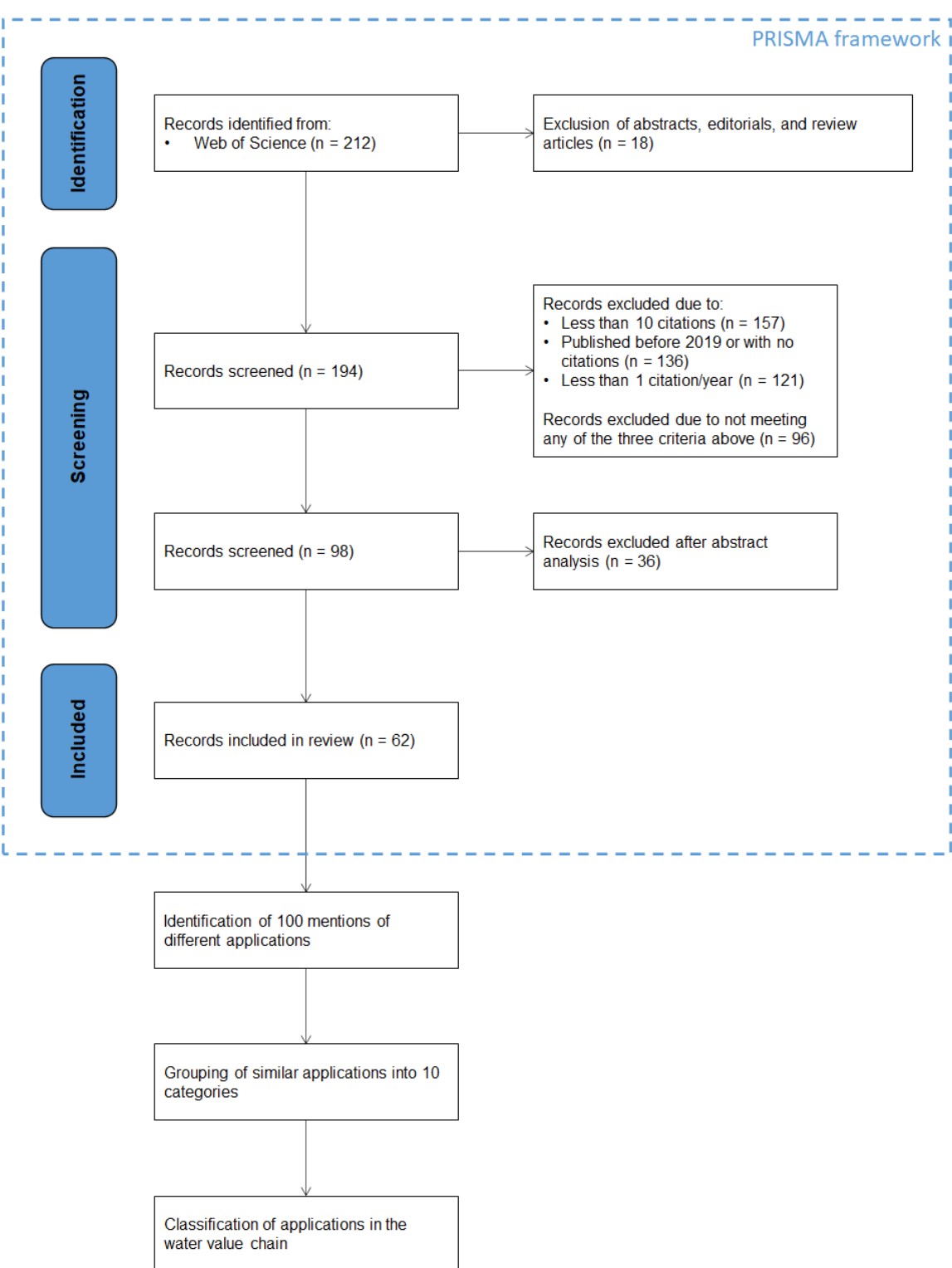

**Figure 3.** Representative flowchart of the methodology steps using the PRISMA framework ([27] adapted by authors).

## 4. Applications of Smart Water Management

The literature review on smart water-management systems aiming to answer the research question Q1 resulted in the identification of 100 mentions of applications in 62 articles, grouped into ten categories. The applications of smart meters stood out, present in 43% of the analyzed articles, followed by implementation models and architectures

(15%) and loss management (11%)—see Figure 4 for the complete ranking of applications. Figure 5 illustrates there has been a significant growth of publications on smart water-management applications in the last five years, driven mainly by the smart-meter category. The ten categories of applications identified were detailed in the following subsections.

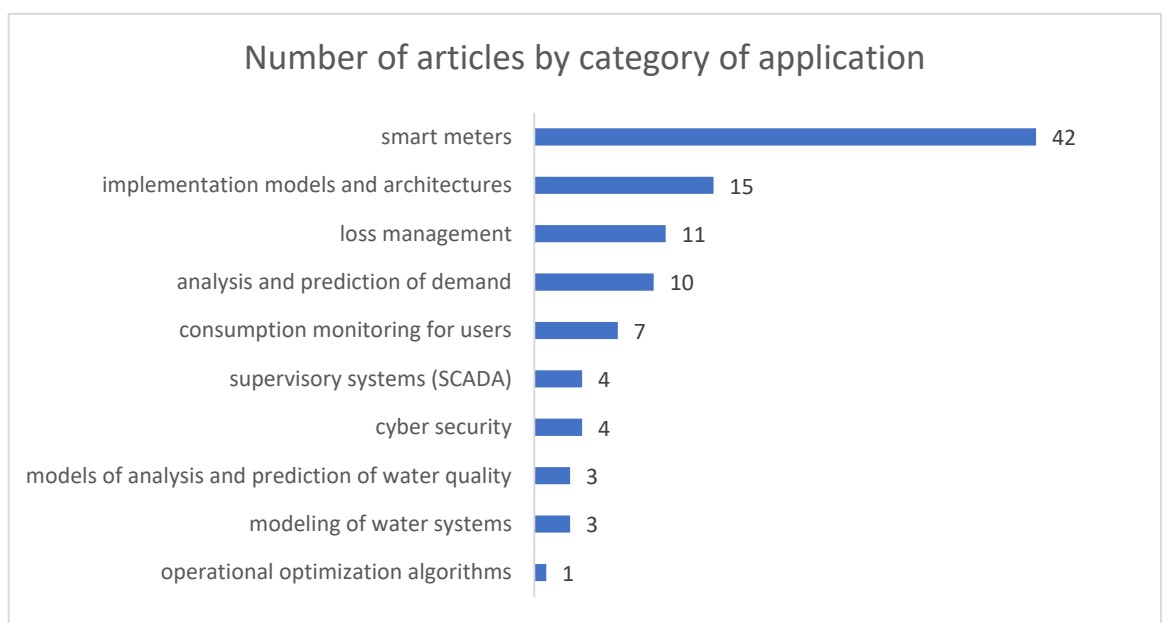

**Figure 4.** Number of articles by category of application (Authors).

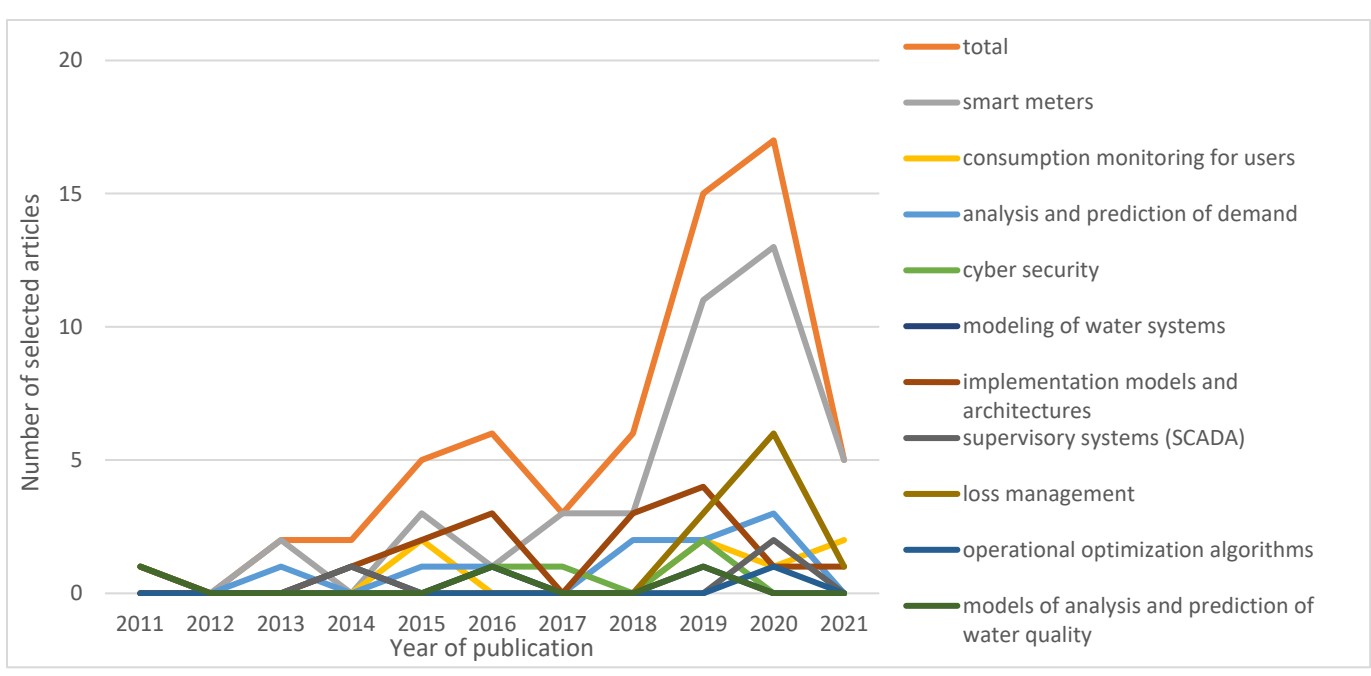

**Figure 5.** Evolution of publications about applications of smart water management over the years (Authors).

### 4.1. Smart Meters

Smart meters made up the main category of smart water-management system applications. Although no formal definition of the concept of smart meters was found, it was possible to conclude from the analyzed articles that their "intelligence" derives from the

ability to record and transmit information about the physical quantity measured through a communication channel. Solutions such as the WM-Bus protocol [28], LoRa and LoRaWAN networked meters with low energy consumption and longer transmission range [22,29–31], and meters based on IEEE 802.15.4 and ContikiOS LibCoAP [32], exemplified the implementation of different communication channels for data transmission. In contrast to these technologies, which presuppose the existence of an electronic meter that generates an electrical signal proportional to the measured variable, Li, C. et al. [13] and Pietrosanto et al. [33] proposed image recognition methods for reading mechanical water flow meters, which is an interesting technology for cities where the installed base of mechanical meters still predominates over electronic ones. All applications found regarding smart meters were listed in Table 1.

**Table 1.** Identified smart meter applications (Authors).

| Applications | References |
| --- | --- |
| Smart meters in general | [1], [5] *, [9,19,21,34–36], [37] *, and [38–50] |
| Arduino and WiFi | [51] * |
| IEEE 802.15.4 and ContikiOS LibCoAP | [32] |
| IoT (LoRa/LoRaWAN, SigFox and NB-IoT) | [22] * and [29,30,52,53] |
| Reading of mechanical flow meters | [13,33] * |
| LinkIt One and Mediatek Cloud Sandbox | [54] |
| Self-powered meters | [14,55] |
| Wastewater flow meter using tomography | [56] |
| Vibration and acoustic meters | [57–60] * |
| Raspberry PI and 6LoWPAN | [31] |
| Raspberry PI and ZR16S08 | [61] * |
| Wireless M-Bus (WM-Bus) | [28] |

* Simultaneous reference to applications in the "smart meters" and "loss management" categories.

In the field of data transmission, Lalle et al. [52] stood out with their comparative study of three technologies of low energy consumption (LPWAN) used in smart water-management systems based on IoT: LoRaWAN, SigFox, and NB-IoT. Their results showed that the NB-IoT had better scalability, i.e., it was capable of aggregating more smart meters at a lower error rate in data transmission packages. The reports on the development of smart meters took into account energy consumption and battery life. For this reason, energy consumption was a restriction when considering the solutions presented by [28,29]. Two strands of work were found aiming at optimizing energy consumption: edge computing architecture and self-powered smart meters. Tzagkarakis et al. [21] demonstrated architectural solutions for smart meters based on edge computing in which smart meters locally processed data compression algorithms for sending them to servers in order to extend battery life. Li, X. J. et al. [14], in their turn, proposed a self-powered water flow meter project based on a turbogenerator to generate a voltage signal proportional to the flow rate in the meter and also generate the energy needed to recharge its battery, together with a wireless signal transmitter and an application for mobile devices to enable users to monitor consumption.

With the development of smart meter technologies over time, smart meters went on from serving only as a tool for measuring consumption and billing customers to becoming part of a series of applications throughout the whole water-management chain. Fabbiano et al. [57], Gong et al. [58], Stephens et al. [59], and Zhang et al. [60] corroborated this argument and proposed the application of vibration and acoustic meters to predict pipe breakage events and leaks, as did Gautam et al. [51], who reported the use of smart meters to collect data from residential reservoir levels, and Jia et al. [34], who dealt with

a new approach to modeling sanitary sewage systems using data from residential smart meters and their geographic location to identify domestic contributions and flows in the sewage network.

The interest of the authorities of Dhaka (Bangladesh) [53] and of Australia and New Zealand [9] in the implementation of smart meters and the good results obtained in reducing water consumption after they were implemented in Capetown (South Africa) [35] were evidence found about the interest in and benefits of such technology. This perception of benefit was corroborated by the volume of scientific production on the subject: 42 articles on smart meter applications. From this set of articles, 15 contextualized smart meters with other applications and, among these, 10 stood out for mentioning smart meter applications and loss management simultaneously (see comments in Table 1).

*4.2. Loss Management*

The International Water Association (IWA) uses the concept of water balance to standardize the terms related to water loss management as described in Table 2. Water losses comprise both those related to leakages (real losses) and to metering issues and unauthorized consumption (apparent losses), which implies lower revenues and also the waste of energy and water as natural resources.

**Table 2.** Standard IWA Water Balance [62].

| | | | |
|---|---|---|---|
| | Authorized consumption | Billed authorized consumption | Billed metered consumption |
| | | | Billed unmetered consumption |
| System input volume | | Unbilled authorized consumption | Unbilled metered consumption |
| | | | Unbilled unmetered consumption |
| | Water losses | Apparent losses | Unauthorized consumption |
| | | | Mering inaccuracies and data handling errors |
| | | Real losses | Leakage on transmission and/or distribution mains |
| | | | Leakage and overflows at the utility's storage tanks |
| | | | Leakage on service connections up to the point of customer metering |

Where "Revenue water" spans Billed metered consumption and Billed unmetered consumption; "Nonrevenue water" spans the remaining rows.

The category of loss-management applications shown in Table 3 constituted one of the main ways in which data generated by smart meters were used in the analyzed articles. Applications focused on real losses management and characteristically comprised collection and treatment of pressure and flow data in the water network to identify leak events, including their locations. The first article selected on the subject was the one dealing with the design, implementation, and operation of TaKaDu, a real-time water-infrastructure monitoring solution in use in Jerusalem (Israel) since 2009 [5]. TaKaDu was responsible for collecting the information that arrived at SCADA, treating it through statistical algorithms, and identifying faults in the network, such as leaks.

**Table 3.** Identified loss-management applications (Authors).

| Applications | References |
|---|---|
| System for loss control | [5,22,33,37,51,61,63] |
| Vibration and acoustic meters for leak identification | [57–60] |

The traditional technique of identifying losses by nighttime minimum flow still proved to be useful, combined with data from smart meters [33,37]. The economic and financial feasibility of applying a loss-management system was proven by [37], who analyzed the

results of a maintenance campaign in the hydraulic installations of 196 schools which used the data from smart meters to identify leaks through this method, obtaining as its main result a higher savings of water than the cost of the maintenance carried out.

Another strand of loss-management applications sought to develop smart vibration and acoustic meters to identify cracks in pipes and, in this way, anticipate leakage events [57–60].

### 4.3. Analysis and Prediction of Demand

The references analyzed (see Table 4) approached demand analysis as the study of information generated from the processing of data sent by smart meters, enabling the evaluation of water-consumption behavior by users [38,44], operational optimization of the water supply system [9], and the identification of consumption patterns and anomalies [45,64].

**Table 4.** Identified applications of analysis and prediction of demand (Authors).

| Applications | References |
|---|---|
| Analysis of demand | [9,38,44,45,64] |
| Models of prediction of demand | [3,19,49,51,65] |

The mass of data provided by smart meters not only allowed the analysis of present demand but also enabled the elaboration of models for predicting future demand. Benítez et al. [19] presented a method of prediction of demand based on similarity techniques that eliminated the need to incorporate aspects of seasonality and allowed predictions within a 24 h horizon. Vijai et al. [3] and Gautam et al. [51] presented solutions based on machine learning, and Vijai et al. [49] supported the application of neural networks.

### 4.4. Consumption Monitoring for Users

This set of studies dealt with software that allowed users to monitor their water consumption online [9,51]. Gautam et al. [51] proposed an IoT-based smart water-management system with software for viewing information about consumption, consumption prediction, and leak detection.

Of the seven articles on consumption monitoring applications for users listed in Table 5, six also mentioned the use of smart meters, corroborating them as a necessary condition for the implementation of such applications. Kounoudes et al. [66] were the exception, as they addressed this issue in the context of user data-privacy preferences.

**Table 5.** Identified applications of consumption monitoring for users (Authors).

| Applications | References |
|---|---|
| Systems in general | [33,41,66] |
| Pandora FMS and web system | [32] |
| Online platform "for users" | [9,14,51] |

### 4.5. Cyber Security

Concern about cyber security in smart water-management systems reported by the articles in Table 6 stemmed from the perception of water scarcity as a threat to the world population and of how cyber attacks could contribute to this scarcity. Considering these issues, Ntuli et al. [67] proposed an architecture for implementing smart water-management system technologies, in which authorization and access-control servers receive authorization requests and grant tokens to devices that produce and consume data. Pritchard et al. [47] discussed different security aspects applicable to software and wireless networks to propose an approach that could combine them into a single solution.

Table 6. Identified cyber-security applications (Authors).

| Applications | References |
|---|---|
| Architecture for security | [47,67] |
| User data privacy | [43,66] |

Within the cyber security topic, a specific strand has developed in recent years to address user data privacy. Kounoudes et al. [66] addressed the issue in the context of a smart home in which water management is prominent, seeking to identify and consider user preferences regarding data privacy. Taking applicable technologies into consideration, Mahmoud et al. [43] presented studies on the implementation of the integration of data from smart meters with blockchain, so as to preserve the identification of users and their consumption data.

### 4.6. Supervisory Systems (SCADA)

Two of the four references to SCADA systems (also known as supervisory systems) listed in Table 7 occurred in the context of integration with other applications, as in the cases already mentioned with real-time simulation models [68] and with TaKaDu [5].

Table 7. Identified applications of supervisory systems (SCADA) (Authors).

| Applications | References |
|---|---|
| SCADA integration with real-time models | [68] |
| SCADA in general | [5] |
| SCADA for sewage management | [12] |
| MAGES | [69] |

Two articles dealt specifically with the use of SCADA in sewage-management processes: Gaska et al. [12] mentioned the adoption of SCADA systems for sewage treatment and wastewater management; Tabuchi et al. [69], in their turn, discussed MAGES, a real-time control system for controlling the pollution caused by rainwater mixed with sewage overflows in the Paris Metropolitan Region (France).

### 4.7. Modeling of Water Systems

The category of modeling of water systems brought together different applications whose objectives were to create computer models to analyze and simulate the real behavior of the systems. The three of them were classified as shown in Table 8.

Table 8. Identified water systems modeling applications (Authors).

| Applications | References |
|---|---|
| Models in real time | [68,70] |
| Models for operational optimization | [40] |

Di Nardo et al. [40] demonstrated how the calibration of a hydraulic model of the water distribution system of Castellammare di Stabia (Italy) was compromised by the use of a few smart meters, which reinforces the benefit of implementing a larger number of them as mentioned above. This paper addressed this application category with the main objective of operational optimization.

Two articles advanced the research area by proposing the adoption of models of water systems in real time. Boulos et al. [68] proposed the integration of supervisory systems (SCADA) with real-time simulation models for more proactive operational actions. Abu-

Mahfouz et al. [70] presented a dynamic hydraulic model integrated with sensors and actuators which showed advantages in comparison with static models.

### 4.8. Models of Analysis and Prediction of Water Quality

The water-quality analysis and prediction models mentioned in the articles in Table 9 aimed to identify events of noncompliance with quality standards, such as contamination. Armon et al. [5] developed TaKaDu, a tool based on the statistical analysis of data, while Vijai et al. [3] presented a solution based on machine learning, and Adedeji et al. [63] demonstrated IoT applications.

**Table 9.** Applications of models of analysis and prediction of water quality identified (Authors).

| Applications | References |
| --- | --- |
| IoT | [63] |
| Machine learning | [3] |
| TaKaDu | [5] |

### 4.9. Operational Optimization Algorithms

The category of applications of operational optimization algorithms comprised the only article selected that deals with the use of data-analysis techniques aimed at improving operational processes in the context of water management (see Table 10). Gaska et al. [12] discussed how to reduce energy consumption by applying artificial intelligence (AI) to optimize processes and make predictive diagnoses in sewage treatment and waste management systems.

**Table 10.** Identified application of operational optimization algorithms (Authors).

| Application | References |
| --- | --- |
| Artificial intelligence (AI) | [12] |

### 4.10. Implementation Models and Architectures

The applications presented above were largely addressed in segmented ways in the articles. The implementation of a "smart water management" system proved to depend on the definition of models which organized different applications. Different solutions were proposed by the articles listed in Table 11. Hauser et al. [71] combined this concern with the issue of interoperability of solutions to propose a structured process for defining system architectures for smart water management. Robles et al. [72] presented a high-level integration model. At a greater level of detail, Gonçalves et al. [7] presented REFlex Water, an architecture for IoT-based smart water management which adopts complex event processing and declarative business process techniques and was applied in a Brazilian municipality. Zhu et al. [50] defined a smart water-management system implementation model based on five levels (smart sensing, infrastructure, water-related big data, smart applications, and access portal) and two subsystems (standardization and regulation and information security).

**Table 11.** Identified applications of implementation models and architectures (Authors).

| Applications | References |
| --- | --- |
| Implementation models and architectures in general | [6,10,15,39,50,71] |
| Big data in Apache Hadoop | [11] |
| IoT-based models | [1,16,47,54,61,73,74] |
| REFlex Water | [7] |

One article focused specifically on the study of methods for treating big data in smart water-management system architectures. Diaconita et al. [11] comparatively analyzed several architectures for processing the big data of smart cities (with water management as one of their application domains), using Apache Hadoop, which they pointed out as a powerful tool for horizontal integration.

## 5. Discussion

This paper adopted a systematic literature review methodology to investigate which smart water-management applications have been mentioned in the scientific literature in a broader way so it could compile and organize the knowledge regarding the subject. It has identified 100 different mentions of applications of smart water-management systems in 62 articles, just what was aimed at by research question Q1. Those applications were grouped into ten categories: smart meters, loss management, analysis and prediction of demand, consumption monitoring for users, cyber security, supervisory systems (SCADA), modeling of water systems, water-quality analysis and prediction models, operational optimization algorithms, and implementation models and architectures. The categories of smart meters and implementation models and architectures stood out, cited in 42 and 15 articles, respectively. This work proved to be more comprehensive than [1], which systematic literature review focused on two categories named water-level monitoring and water-quality monitoring. The set of application categories defined in this article composes a wider framework for smart water management.

It is important to emphasize that the manuscripts reviewed in this study do not fully describe the ten categories here defined, so some indepth research regarding each of them may be valuable for a deeper understanding of smart water management. One example is earlier research regarding the adoption of AI in 49 large water utilities in the United States, finding that: (a) 24% of them had used some kind of AI; (b) more than half were planning to do so; (c) saving money appeared as the main driver for AI implementations, followed by leak detection and water-quality improvement; (d) the main concerns regarding its implementation were uncertainty regarding the return of investments and lack of AI knowledge [75]. Therefore, in spite of not having any of the keywords listed on the research string defined in our manuscript; this manuscript could fit "loss management", "models of analysis and prediction of water quality", and "operational optimization algorithms" categories.

The recurrence of the subject of smart meters was justified by the fact that they constitute the basis of the architecture of smart water-management systems, given their ability to transmit data on the measured physical quantities which serve as a source for the other applications. The strong correlation between this application category and the loss-management category indicates the first steps towards the implementation of a smart water-management system and can be detailed in a future technological roadmap study together with the major framework composed of all ten categories described in this article, in addition to some investment and cost analysis of its implementation.

Loss-management applications deserve a highlight due to their inherent relation to more sustainable and efficient use of water resources, as demonstrated by applications mentioned in the referred section of this article. Thus, this application category plays a fundamental role in the pillars of the smart water-management concept presented in this manuscript.

The analysis of the coverage of applications in the water value chain aiming to answer the research question Q2 resulted in a predominance of applications in water distribution and utilization processes and, also, in water-management systems as a whole (see Figure 6). The volume of scientific production on smart meters and loss management, topics closely related to the processes in question, contributed to this predominance. On the other hand, the total absence of applications aimed at water treatment and effluent reuse processes was highlighted, even though they are pointed out as adding value to users [25]. Therefore, the

need to develop more indepth studies on possible existing applications or the development of new ones for these processes was justified.

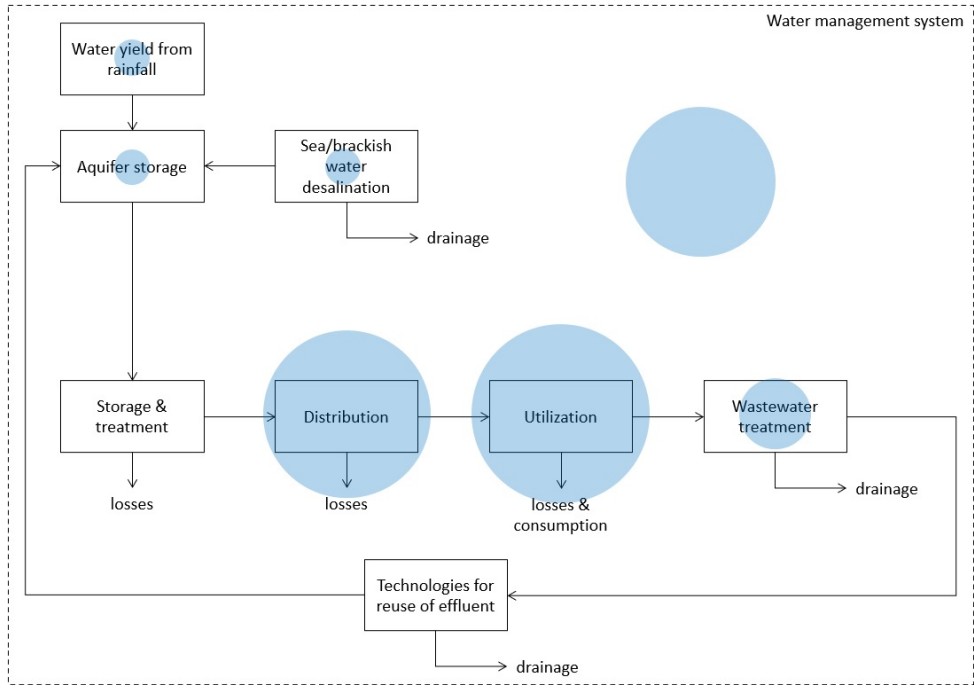

**Figure 6.** Coverage of smart water system applications in the water value chain (adapted from [26]).

Finally, the survey of applications of smart water-management systems allowed a better understanding of their meaning in more concrete terms. Throughout the description of the applications, interrelationships between some categories were highlighted, with emphasis on smart meters together with loss management and consumption monitoring for users. Although the articles presented integration models for smart water-management systems, they did not specify how different applications would fit into different hierarchical levels. Consequently, the development of an integration model or framework for the applications of smart water-management systems identified in this article could contribute towards the development of smart water-management systems.

**Author Contributions:** Conceptualization, methodology, and data collection were performed in co-operation by É.S.A. and F.M.M. Data analysis was independently performed by É.S.A. and F.M.M. The first draft of the manuscript was written by É.S.A. and all authors commented on previous versions of the manuscript. Research guidance, discussion, and critical review were made by C.F.M.A., N.K. and E.M.D. All authors have read and agreed to the published version of the manuscript.

**Funding:** This study was financed in part by the Coordenação de Aperfeiçoamento de Pessoal de Nível Superior—Brasil (CAPES)—Finance Code 001.

**Data Availability Statement:** The data presented in this study are available in all the references listed.

**Conflicts of Interest:** The authors declare no conflict of interest.

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
