# Peer review of "Applications of Smart Water Management Systems: A Literature Review"

_water, doi:10.3390/w15193492_

Round 1

Reviewer 1 Report

This could be a valuable review. My main concern is that the authors’ focus on “smart” terminology misses key articles that don’t exactly say “smart” but that still meet the intent, so the review is therefore incomplete and unrepresentative. If they can remedy this weakness I will reevaluate the article.

Lines 53 (and 70, 75, 93, 94 and others): A numbered reference should not be the subject of an active sentence. Instead, say, “Smith [19] defined this concept …”

Line 59: The focus on academic literature is appropriate and necessary. However, I wonder how this diverges from actual practice? Researchers, by definition, develop new technologies, while practitioners (actual water systems) may lag behind, so the academic literature may not represent real uptake of smart technologies.

General: Rapp et al. (https://doi.org/10.1061/JWRMD5.WRENG-5870) recently surveyed AI applications among drinking water utilities. It is highly relevant and supports some of the authors’ conclusions.

Line 61: What is meant by value chain?

Lines 75–89: In this discussion of layers, I would suggest there is a policy layer too: water rights, health regulations, facility permits, etc.

Line 126 and Figure 3: The PRISMA method needs a citation.

Figure 4: I like the categorization, which makes it easy to grasp the applications.

Figures 4 and 5: Please increase the resolution.

Figure 4: The 8th category truncated.

General: The finding of few articles categorized as “operational optimization algorithms” (and other categories) is perhaps a result of the search approach, which lacked any strings for those terms. Pump scheduling and real-time control, for example, are big ones that are well studied. See, for example, Creaco et al. (https://doi.org/10.1016/j.watres.2019.06.025) and Luna et al. (https://doi.org/10.1016/j.jclepro.2018.12.190). The authors should reconsider their search specifically for this category, and possibly the water quality modeling category. There are dozens more relevant articles, even if they do not explicitly use the word “smart.”

Figure 5: Is the decline in 2021 real, or was the database perhaps incomplete at the time of the authors’ query?

Table 1: Footnote should appear below table.

Section 4.2: Rapp et al. confirm that leak detection is a popular use of AI.

Table 2: Needs a citation to IWA/AWWA.

Section 4.9 and Table 10: An opportunity to discuss Rapp et al. (for more AI), Creaco et al., and many others. This is a major weakness in the authors’ analysis.

Minor.

Reviewer 2 Report

The means by which water distribution systems are managed within the water supply companies and the skillful management of collected data is of increasing importance in the context of reducing water losses and sustainable management of water resources. However, the presented literature does not provide significant information regarding the implementation of tested data and the current state of knowledge. In my opinion, the way in which the issue has been presented should be revised.

The authors have presented a solution for water system management. However, the authors have not sufficiently presented the advantages and disadvantages of these methods. There is no information regarding the applications of these particular methods in the distribution system and no information regarding profits and cost of their implementation. The factors which are limiting their implementation should also be included.

The data collection and transfer should be presented in a more detailed manner, for example; what type of data would be useful in water system management and what are the benefits of their implementation?  The methods of interpretation, analysis of collected data, and practical aspects of their implementation should also be presented, together with modern methods of water system management including hydraulic and quality parameter analysis and the results of their application.

The aim of the reviewed article is unclear and the presentation of the research is useless with regards to making a decision on which management method is more beneficial.

Round 2

Reviewer 1 Report

I am satisfied with the revision.

Author Response

Thanks for your comments.

Reviewer 2 Report

Stil, it is not clear. What type of data would be useful in water systems managers and what benefits from its implementation?  The methods of interpretation and analysis of collected data and practical aspects of  its implementation should be presented. Modern methods of water system managers including hydraulic and quality parameters analysis should be presented and its application results.

Presented literature review is similar statistics analysis not scientific assess of state of art in water system managers and presentation of  managers tool or its effects in decrease of water loss or effectiveness of data management.

The aim of literature review is not clear and presentation of research are useless in make decision in managements method choose.
